# Sparse Representation Graph for Hyperspectral Image Classification Assisted by Class Adjusted Spatial Distance

**Wanghao Xu [1], Siqi Luo [1], Yunfei Wang [1], Youqiang Zhang [2] and Guo Cao [1],***

[1]  School of Computer Science and Engineering, Nanjing University of Science and Technology, Nanjing 210094, China; xu.wanghao@njust.edu.cn (W.X.); siqiluo@njust.edu.cn (S.L.); woilf@njust.edu.cn (Y.W.)

[2]  School of Internet of Things, Nanjing University of Posts and Telecommunications, Nanjing 210023, China; yq_zhang@njust.edu.cn

*  Correspondence: caoguo@njust.edu.cn

**Abstract:** In the past few years, the sparse representation (SR) graph-based semi-supervised learning (SSL) has drawn a lot of attention for its impressive performance in hyperspectral image classification with small numbers of training samples. Among these methods, the probabilistic class structure regularized sparse representation (PCSSR) approach, which introduces the probabilistic relationship between samples into the SR process, has shown its superiority over state-of-the-art approaches. However, this category of classification methods only apply another SR process to generate the probabilistic relationship, which focuses only on the spectral information but fails to utilize the spatial information. In this paper, we propose using the class adjusted spatial distance (CASD) to measure the distance between each two samples. We incorporate the proposed a CASD-based distance information into PCSSR mode to further increase the discriminability of original PCSSR approach. The proposed method considers not only the spectral information but also the spatial information of the hyperspectral data, consequently leading to significant performance improvement. Experimental results on different datasets demonstrate that compared with state-of-the-start classification models, the proposed method achieves the highest overall accuracies of 99.71%, 97.13%, and 97.07% on Botswana (BOT), Kennedy Space Center (KSC) and the truncated Indian Pines (PINE) datasets, respectively, with a small number of training samples selected from each class.

**Keywords:** hyperspectral image classification; semi-supervised learning; sparse representation; spatial distance information; regularizer

## 1. Introduction

A hyperspectral image (HSI) records a wide range of electromagnetic wave data reflected by the earth's surface. HSI has been widely used in agricultural mapping [1] and mineral identification [2], and due to its high-resolution spectral record of the land covers, HSI data is suitable for the classification of different objects on land [3–5]. However, among all HSI data acquired, the labeled one is very limited. In this situation, semi-supervised learning (SSL) provides a promising way to deal with both the limited labeled data and the rich unlabeled data [6,7].

In recent years, many groups have applied SSL methods to the HSI classification area. The typical SSL methods include the self-training method [8], the collaborative training method [9], the generative model method [10] and the graph-based method [11]. The self-training method [8] adds pseudo-labels to high-confidence unlabeled samples in each iteration until all the unlabeled samples are labeled. The collaborative learning [9] is proposed to make the HSI classification performances more reasonable

and promising within limited labeled data samples which combines activate learning (AL) with SSL. The generative models such as expectation-maximization algorithms with finite-mixture models [10] have been applied for HSI classification. It is worth mentioning a self-training method based on convolutional neural networks (CNN) proposed by Wu et al. [12]. In their work, authors propose a CNN-based classification framework which uses self-training to gradually assign pseudo labels to unlabeled samples by clustering and employs spatial constraints to regulate self-training process. It is an attractive work that combines the spectral neighborhood information with the spectral information and achieves high performance on several datasets. However, the CNN-based method could be time-consuming at the training stage, and the performance of a self-training model is highly dependent on the initial samples it chooses.

Among all SSL methods, the graph-based method [13] has attracted attention from many researchers because it is easy to analyze the mathematical formulation and can obtain a close-form solution. On the other hand, sparse representation (SR) provides us a reliable way, due to its solid foundation in mathematics, to describe the linkage between samples, which could help the graph building. The SR method was first introduced by Yan et al. [13] and Cheng et al. [14] to generate the L1-graph. Afterwards, the SR-based graph method was applied in the HSI classification [13–17]. For example, Gu et al. [15] proposed the L1-graph semi-supervised learning model for hyperspectral image classification, and Shao et al. [17] presented the probabilistic class structure regularized sparse representation (PCSSR) approach which outperforms state-of-the-art algorithms in graph construction in most cases [18,19]. Different from normal SR methods, the PCSSR approach introduces a probabilistic class structure regularizer into the SR model, where probabilistic class structure reflects the probabilistic relationship between each sample and each class, and further, calculates the distance between each two samples based on their probabilistic relationship. With the distance information provided, the process of the SR algorithm will be guided by it. Therefore, the key point of the PCSSR algorithm is the distance information and how to generate it properly. In the previous study, however, researchers only apply another SR process to compute out the distance information, which only focuses on the spectral information but fails to utilize the spatial information.

Despite the highly discriminative capability to achieve high classification accuracy, PCSSR suffers from the limitation for neglecting the spatial information of HSI. Since sample pixels have the characteristics of spatial continuality, failing to consider spatial information would miss such important characteristics that are beneficial for enhancing classification capability. Referring to the classification results in PCSSR paper, within a wide range of land covers for a certain class, we may observe mislabeled pixels that have been classified into a wrong class. Therefore, we conclude that the classification results by using only spectral information would lack spatial continuality and smoothness.

In order to address the above-mentioned limitation, this work aims to incorporate the spatial distance information into PCSSR to improve the discriminative capability of PCSSR. In addition, for better estimating the spatial distance, we propose a new measurement method for spatial distance called class adjusted spatial distance (CASD). This new method takes into account both the spatial distance and class difference between each two pixels. By such means, we can obtain appropriate discriminative information for pixels belonging to the same class but with long spatial distance, by assigning a relatively small CASD value. The effectiveness of employing CASD for the regularization process in PCSSR was thoroughly verified by the experimental results. Experimental results on different datasets demonstrate that the proposed method can significantly improve the classification accuracy by incorporating the spatial information in the CASD metrics. Compared with state-of-the-start classification models, the proposed method achieves the highest overall accuracies of 99.71%, 97.13%, and 97.07% on BOT, KSC, and truncated IND PINE datasets, respectively, with a small number of training samples selected from each class. Specifically, the main contributions of this paper include the following two aspects.

1.  We propose the concept of the CASD. The calculation of the CASD based mainly on the planar Euclidean distance and the shortest path algorithm. The CASD takes the class similarity between samples into consideration, which can make the measurement of distance more accurate and reasonable.
2.  We apply the CASD to estimate the distance information needed in the PCSSR algorithm. The results show that, this approach can enhance the performance of the PCSSR algorithm when enough training samples are provided. We achieve the highest improvement of classification accuracy of 8.65% and 3.85% on the KSC and the BOT dataset when the number of labeled samples selected from each class reaches 20, and achieve 15.97% on the truncated IND PINE dataset when the number of labeled samples selected from each class reaches 15.

## 2. Related Works

This section provides a brief discussion of existing graph construction methods for HSI classification. During the process of graph-based SSL method, label propagation (LP) is a crucial step for transferring labels from a limited number of labeled samples to abundant unlabeled samples [6] with a given graph which denotes the connection among all samples. The basic idea of the LP algorithm is to assume that similar samples should have similar labels, so the mathematical way of achieving this purpose is to define an energy function (see Equation (8)) for the given graph that is used to judge the "smoothness" of the classification results—if the results meet the assumption of LP (i.e., similar samples should have similar labels), the value of the energy function will be small and vice versa.

To implement the above-mentioned procedure, we need to first obtain a well-constructed graph and provide an accurate adjacent matrix. The adjacency matrix of the graph reflects the relationship between samples, and a well-constructed graph should denote the similarity between samples honestly. Therefore, we need to find a good and proper method to generate an accurate similarity matrix, i.e., the adjacency matrix of the graph. Different from traditional graph construction methods, SR-based methods have the capabilities of learning the local relationship from samples and computing the well-discriminated edge weights of the graph, and therefore are robust to noises and parameter variations. We discuss below some representative methods in these two categories.

### 2.1. Traditional Graph Construction Methods

The process of graph construction is momentous in graph-based SSL which mainly involves two steps: building the graph adjacency structure and calculating the graph edge weight. For building graph adjacency structure, k-nearest neighbors (KNN) and $\varepsilon$-ball neighborhood are the two most popular approaches [20]. As for graph weighting methods, Zhou et al. [21] use the Gaussian kernel (GK) function to calculate the edge weight, however if only a few labeled samples are provided, it will be hard to determine the hyper-parameters in the function [22]. Wang et al. [22] propose a non-negative local linear reconstruction (LLR) to use the neighborhood information of each data point to construct a graph in order to derive a more reliable and stable way to construct the graph. First, they approximate the entire graph as a series of overlapping linear neighborhood patches, then they find the edge weight of each linear neighborhood patch, and then they aggregate all the edge weights together to form the edge weight matrix of the entire graph; Ma et al. [23] consider that sparsity is essential for improving the efficiency of SSL algorithms. Therefore, they propose local linear embedding (LLE)-based weight which can capture the local geometric properties of hyperspectral data and is good for weighting the graph edge in a low-level computational cost. Zhuang et al. [24] proposed nonnegative low-rank and sparse (NNLRS) approach to use both low-rankness of high dimensional data samples and the sparsity to construct a good graph. The obtained graph can capture the local low-dimensional linear structures of the data samples and the global cluster or subspace structures of the data samples.

However, these traditional methods share the same disadvantages that they all have fixed manually tuning parameters. As a result, this category of graph construction methods are very sensitive to the data noise and parameter variations.

## 2.2. SR-Based Graph Construction Methods

Unlike the traditional graph generation approaches, the SR-based methods can learn the local relationship from samples and compute the well-discriminated edge weights of the graph. By encoding a certain sample as a sparse linear combination of all the other samples, the sparse coefficients of the linear combination can be viewed as the edge weights from the certain sample to all the other samples [13,14]. By doing so, the graph that LP algorithm demanded could be generated.

In addition to the most SR based methods, Shao et al. proposed the probabilistic class structure regularized sparse representation (PCSSR) approach. In their work, the authors manage to incorporate the SR model with a probabilistic class structure that reflects the probabilistic relationship between each sample and each class. Further, with the probabilistic class structure provided, the distance between each two samples can be acquired according to the difference between their probabilistic class labels. Finally, a class structure regularization is developed using the distance between each two samples. The authors claim that, with the class structure regularizer, PCSSR can learn a more discriminative graph from the data, and as shown in the experimental results, the PCSSR method outperforms state of the art on Hyperion and airborne visible infrared imaging spectrometer (AVIRIS) hyperspectral data. The class structure regularizer and the full model of PCSSR are shown in Equations (1) and (2), respectively, where $W$ is the adjacency matrix we need to obtain for the LP algorithm, $M$ is the distance matrix and each entry $M_{ij}$ represents the distance between two samples based on the difference between their probabilistic class label, and $X$ denotes all samples in training set and testing set.

$$R(W) = \sum_{i,j} |W_{ij} \cdot M_{ij}| \tag{1}$$

$$\min_{W} \frac{1}{2} \|X - XW\|_F^2 + \lambda_1 \|W\|_1 + \lambda_2 R(W) \text{ s.t. } diag(W) = 0, \ W \geq 0, \tag{2}$$

However, the probabilistic class structure used in the PCSSR paper in obtained only through another SR process, which fails to take into account the abundant spatial information in the HSI dataset. Despite the highly discriminative capability to achieve high classification accuracy, PCSSR suffers from the limitation for neglecting the spatial information of HSI. Since sample pixels have the characteristics of spatial continuality, failing to consider spatial information would miss such important characteristics that are beneficial for enhancing classification capability. Therefore, we conclude that the classification results by using only spectral information would lack spatial continuality and smoothness. In order to address the above-mentioned limitation, our work aims to incorporate the spatial distance information into PCSSR to improve the discriminative capability of PCSSR, which will be introduced and tested in the following sections.

## 3. Modeling and Algorithms

This section details the proposed HSI classification approach that introduces CASD in a SR graph-based method, in order to take advantage of spatial information for improving the classification accuracy. The fundamental idea is to use our proposed CASD instead of the distance matrix $M$ acquired by SR process in the original PCSSR method to measure the distance between any two samples. The CASD-assisted PCSSR can achieve a more accurate and reasonable measurement of sample distances. We further employ the LP algorithm to predict the probability of each unlabeled pixel belonging to a certain class. Figure 1 illustrates the general flow of the proposed CASD-assisted HSI classification method. In what follows, we describe in detail the main steps in this classification flow.

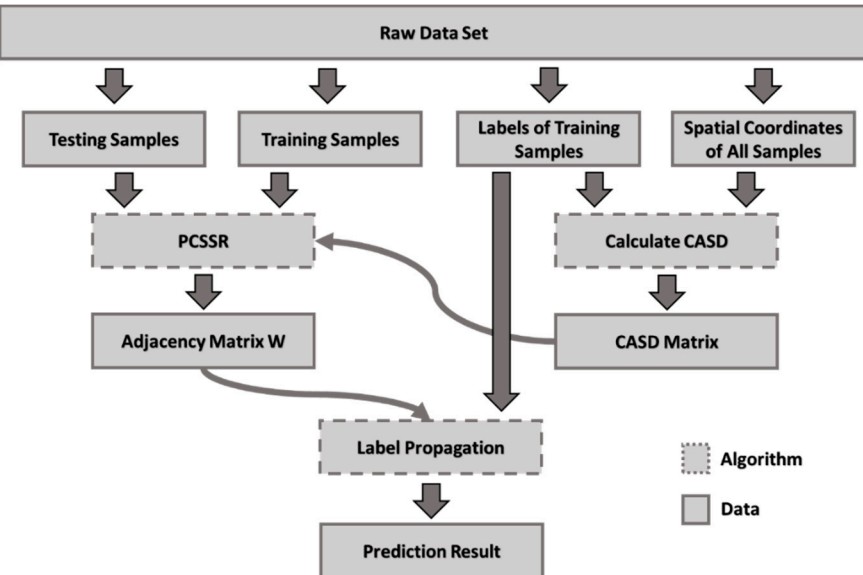

**Figure 1.** The general flow of the proposed class adjusted spatial distance (CASD)-assisted hyperspectral image (HSI)_classification method.

### 3.1. Class Adjusted Spatial Distance

For the purpose of incorporating spatial information into PCSSR, we propose using CASD to replace distance matrix $M$ required by SR process in the original PCSSR. We first provide a brief introduction to th planar Euclidean distance (PED). Consider two points $(a_1, b_1)$, $(a_2, b_2)$ in a plane. The PED between these two points is defined as:

$$d = \sqrt{(a_1 - a_2)^2 + (b_1 - b_2)^2} \tag{3}$$

As we have discussed, to improve the performance of the PCSSR algorithm, a proper distance measurement between each two samples is needed. The distance matrix should reflect the similarity or difference among samples. Since each sample is just an area on the ground, the simplest way to measure the distance between each two samples is by calculating the spatial distance, i.e., the PED between them. The distribution of land covers is usually in a continuous way, so if a sample belongs to some class $c_i \in \{c_1, c_2, c_3, \ldots\}$, the samples in its spatial neighborhood are likely to belong to the same class as it. Thus, we can use the PED between two samples to represent their similarity.

However, PED has its limitation for measuring the distance information that PCSSR needs. It is possible that two samples distant from each other belong to the same class, which is not unusual in the land cover classification. In this case, PCSSR using PED would fail to classify such samples. To overcome this limitation, we introduce the class adjusted spatial distance (CASD) to replace the naïve planar Euclidean distance. Generally speaking, the CASD is a distance measurement which considers not only the Euclidean distance between two samples but also their class difference. We mainly use the Euclidean distance algorithm and the shortest path algorithm to solve the CASD.

We first generate a complete undirected graph $G(V, E)$ where $V$ represents all the $n$ samples and $E$ is valued with the Euclidean distances between every two samples. The distance from a sample point to itself is defined as 0. Then, we check all the labeled samples (vertices) in the complete graph $G$. If two labeled samples belong to the same class, we change the edge weight between them to 0. In this way, we make the samples with the same class "closer" to each other. At the last step, we apply the shortest path algorithm (for example Dijkstra algorithm [25]) between every two vertices in the graph $G$ and revalue the edge weight between them with the length of the computed shortest path. We define this new edge weight as "the class adjusted spatial distance". The above process is illustrated in Figure 2, and the Algorithm 1 is described below.

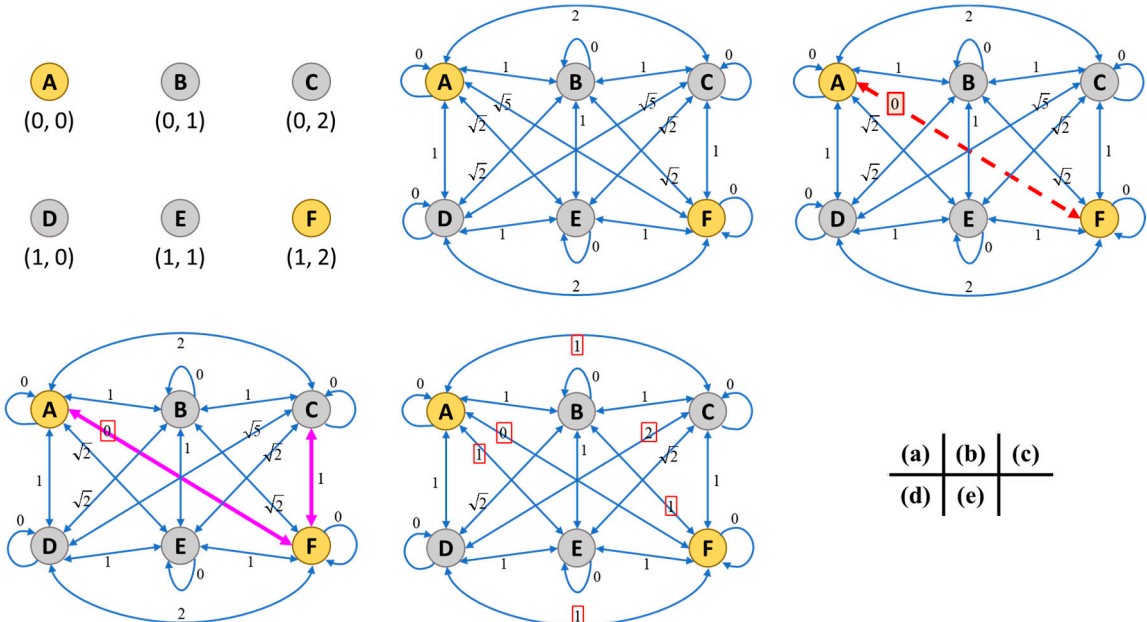

**Figure 2.** A graphical illustration of the CASD algorithm. (**a**) The raw dataset with A~F six samples, where A, F are the labeled samples with the same class and the rest are unlabeled samples to be predicted. The subscript below each sample shows its pixel location in the hyperspectral data. (**b**) Construct a complete undirected graph where each vertex represents a sample and the edge between every two samples is weighted by their Euclidean distance. (**c**) A and F are the labeled samples with the same class, so reweight the edge between them by zero. (**d**) For every two vertices, compute the shortest path between them (the shortest path between A and C is marked in magenta). (**e**) Update the weight between every two vertices with the length of the shortest path between them. The new edge weight is called "the class adjusted spatial distance".

---

**Algorithm 1:** Compute CASD for each two samples

---

**Input:** Array with the spatial coordinates of $l$ labeled samples $Cord_l = [(a_1, b_1), (a_2, b_2), \ldots, (a_l, b_l)]$ and the coordinates of $u$ unlabeled samples $Cord_u = [(a_{l+1}, b_{l+1}), (a_{l+2}, b_{l+2}), \ldots, (a_{l+u}, b_{l+u})]$, the label vector that notes the class of every labeled sample $Label = [c_1, c_2, \ldots, c_l]$

---

**Output:** The adjacency matrix $M \in \mathbb{R}^{n \times n}$, $n = l + u$

1. Weight the edges in the graph by:
   $M(i, j) = EuclideanDistance(Cord(i), Cord(j))$

2. Update the edge weight $M(l_1, l_2)$ between every two labeled samples $l_1, l_2$ according to the following equation:
   $$M(l_1, l_2) = \begin{cases} 0 & , Label(l_1) = Label(l_2) \\ M(l_1, l_2) & , \text{otherwise} \end{cases}$$

3. Calculate the shortest path between every two vertices $v_1, v_2$ in the graph:
   $Path(v_1, v_2) = ShortestPathAlgo(v_1, v_2)$

4. Update the edge weight $M(v_1, v_2) = length(Path(v_1, v_2))$

---

The element value $M_{ij}$ in the output adjacency matrix $M$ represents the calculated CASD between the i-th sample and j-th sample.

## 3.2. CASD-Assisted PCSSR

Based upon the CASD metric defined in Section 3.1, we now describe how to generate the graph for the LP algorithm by using the PCSSR flow. To start with, the PCSSR-based graph generation method is derived from the typical SR-based method. For every sample, the SR based method aims to encode

it as a sparse linear combination of the other samples [13,14]. The typical SR model is formulated as follows:

$$W = argmin\|W\|_1 \text{ s.t. } X = XW, diag(W) = 0, \ W \geq 0, \tag{4}$$

where $X$ denotes all the samples in training set and testing set; $\|\cdot\|$ represents the L-1 norm. By solving this regularization model, we can obtain the graph weight matrix $W$ demanded in the following LP process.

Furthermore, due to the complex working environment and contamination during the data transmission, many hyperspectral images are corrupted by different types and amounts of noises, two common types of which are stripping noise and salt-and-pepper noise. Therefore, considering the corrupted data and the noise during collection, the method can be rewritten as follow to enhance the robustness against noises:

$$W = argmin\frac{1}{2}\|X - XW\|_F^2 + \lambda\|W\|_1 \text{ s.t. } diag(W) = 0, \ W \geq 0, \tag{5}$$

where $X$ denotes all the samples in training set and testing set and $\lambda$ is a tradeoff parameter that controls the sparsity of $W$.

In the next step, we come to a point of divergence from the original paper—the original PCSSR paper next introduces a probabilistic class structure term $P = [P_l; P_u] \in \mathbb{R}^{n \times c}$ where $P_{ij}$ represents the possibility that a sample $i$ belongs to the class $j$, and then calculates the distance matrix $M$ based on the probabilistic class structure $P$, where $M_{ij} = \frac{1}{2}\|P_i - P_j\|^2$. It is necessary to state that, in the original PCSSR paper, the probabilistic class structure $P$ is generated through a standard SR process, and one of the aims of our work is to introduce the spatial information into the PCSSR.

Therefore, instead of computing the probabilistic class structure, we run Algorithm 1, as proposed in Section 3.1, to get the CASD information between each two samples and apply the CASD information as the new distance matrix $M$, where $M_{ij}$ measures the distance between the i-th and j-th sample. If the two samples are close to each other (by category or in spatial), $M_{ij}$ will be a small number, which means they are similar to each other. The additional regularizer for graph edge matrix $W$ is as follows:

$$R(W) = \sum_{i,j}\left|W_{ij} \cdot M_{ij}\right| \tag{6}$$

Obviously, to acquire a smaller $R(W)$, $W_{ij}$ needs to be small when $M_{ij}$ is a large number. By this regularizer, the linkage between two far-away samples will be regularized into a weak linkage. Once we obtain the similarity (or the distance) matrix $M$ by calculating CASD, the final formula of our CASD-assisted PCSSR approach is formulated as:

$$\min_W \frac{1}{2}\|X - XW\|_F^2 + \lambda_1\|W\|_1 + \lambda_2 R(W) \text{ s.t. } diag(W) = 0, \ W \geq 0, \tag{7}$$

where $\lambda_1$ controls the sparsity of $W$, $\lambda_2$ controls the effect of class structure regularizer. The model formulated in (7) is a constrained optimization problem and can be relaxed and solved by Lagrange multipliers methods, for example the alternating direction methods of multipliers (ADM) [26]. However, ADM has the disadvantage of introducing extra variables and requiring parameter tuning. In this work, following the original PCSSR method [17], we employ the ADM with adaptive penalty (ADMAP) [27], which can overcome the above-mentioned limitations, to solve problem (7).

### 3.3. Label Propagation

After getting the sparse graph and its adjacency matrix $W$, we can obtain the final prediction result by using the LP algorithm on the obtained graph. As mentioned in Section 2, the main purpose of the LP algorithm is to transfer labels from the labeled samples to unlabeled samples, and during this process, a prediction matrix will be generated. Furthermore, the generated prediction results

should meet the basic assumption of LP algorithm that similar samples should have similar labels. The mathematical way of achieving the purpose of the LP algorithm is to define an energy function $E(f)$ with a given graph and to minimize the function $E(f)$.

$$E(f) \;=\; \frac{1}{2}\sum_{i,j} W_{ij}\|f_i - f_j\|^2 \tag{8}$$

where $f_i, f_j$ are respectively the predicted label vectors of the i-th and j-th data samples. $f$ is composed of all the predicted label vectors. The matrix $W$ is the adjacency matrix of the graph needed for the LP process.

In order to maintain the experimental consistency with the original PCSSR paper, we follow the formula of LP algorithm used in the original PCSSR paper. The full explanation and the adapted formula are detailed as follows.

The labeled samples are expressed as $X_l = [x_1, x_2, \ldots, x_l]$, and a large number of unlabeled samples $X_u = [x_{l+1}, x_{l+2}, \ldots, x_{l+u}]$. There are total $C$ classes denoted as $C = \{1, 2, \ldots, c\}$. Let $n = l + u$ be the total number of data samples, and usually, the value $l$ is much smaller than $u$. The matrix $W \in \mathbb{R}^{n \times n}$, the adjacency matrix of graph $G$ which can be obtained from the PCSSR process, implies the similarity or the connection between each two samples. Next, we define a label matrix $Y_l$ with $l$ rows, where each row $Y_{li} \in \mathbb{R}^{1 \times c}$ is a one-hot vector representing the class that the corresponding labeled sample $x_i$ belongs to. $F \in \mathbb{R}^{n \times c}$ is the prediction matrix, of which each element $F_{ij}$ represents the probability of the i-th sample belonging to the j-th class; $F_l \in \mathbb{R}^{l \times c}$ is the upper $l$ rows of $F$, while $F_u \in \mathbb{R}^{u \times c}$ is the lower $u$ rows of $F$.

$$\min_{F \in \mathbb{R}^{n \times c}} \frac{1}{2}\sum_{i,j=1}^{N} W_{ij}\|f_i - f_j\|^2 = Tr\!\left(F^T L_W F\right) \text{ s.t. } F_l = Y_l, \tag{9}$$

where the expression after *min* can be viewed as the energy function; $f_i \in \mathbb{R}^{1 \times c}, f_j \in \mathbb{R}^{1 \times c}$ are the predicted label vector of the data sample $x_i, x_j$. $L_w = D - W$ is the Laplacian matrix where $D$ is a diagonal matrix, and $D_{ii} = \sum_j W_{ij}$.

Then we split $L_W$ into 4 blocks by the number of labeled and unlabeled samples:

$$\begin{pmatrix} L_{W_{ll}} & L_{W_{lu}} \\ L_{W_{ul}} & L_{W_{uu}} \end{pmatrix}$$

Finally, we get the prediction matrix that records the possibility of each unlabeled sample belonging to each class:

$$F_u = -L_{W_{uu}}^{-1} L_{W_{ul}} Y_l, \tag{10}$$

The final prediction result for every unlabeled sample is given by:

$$y_i = \arg\max_{j=1,2,\ldots,c} F_u(i, j), i = 1, 2, \ldots, u \tag{11}$$

where $y_i$ denotes the class that the unlabeled sample $i$ is most likely to belong to.

## 4. Experimental Results and Analysis

In this section, we will test the CASD assisted PCSSR algorithm on six different hyperspectral datasets. The algorithm is implemented with MATLAB 2019b and runs on a laptop with i5-7300HQ and GTX 1050TI. We use traditional graph-based algorithms in comparison. The codes and datasets used to generate the results and figures are available in Code Ocean [28].

### 4.1. Experimental Datasets

Two groups of datasets are used to evaluate our model. The first group includes the whole Botswana (BOT) dataset, the whole Kennedy Space Center (KSC) dataset, and the truncated Indian

Pines (truncated IND PINE) dataset where the labeled ground blocks are relatively discrete and distant from each other. Different from the first group, the labeled samples in the second group are less discrete and always appear in bulk. The whole Indian Pines (IND PINE) dataset, the whole Salinas (SAL) dataset, and the whole Pavia University (PAV) dataset are included.

The BOT dataset was collected by the Hyperion sensor on EO-1 satellite over the Okavango Delta, Botswana in May 2001. The 242 spectral bands of the Hyperion image are ranging from 357 to 2576 nm with a spatial resolution of 30 m. Total number of 145 bands in BOT are left after removing some un-calibrated and noisy bands. The KSC, IND PINE and SAL dataset were separately gathered by the AVIRIS sensor over the Kennedy Space Center on March 23, 1996, over the Indian Pines test site in North-western Indian in 1992 and over the Salinas Valley, California, with 224 spectral reflectance bands in the wavelength ranging from 400 to 2500 nm. The un-calibrated bands and noisy bands covering the water absorption feature are removed and only 200 bands remain. The PAV dataset was acquired by the reflective optics system imaging spectrometer (ROSIS) sensor over Pavia University with 103 spectral bands and a spatial resolution of 1.3 m. Before analysis, some of the samples which contain no information are discarded.

More information of these six datasets can be found in [29], and all datasets can be downloaded from [28]. The ground truth of every dataset is shown in Figures 3 and 4. The sample size of each class in each dataset is shown in Tables 1 and 2.

**Table 1.** Sample size (number of pixels) of each class in datasets from Group I.

| Class No. | Botswana (BOT) | | Kennedy Space Center (KSC) | | Truncated Indiana Pine (Truncated IND PINE) | |
|---|---|---|---|---|---|---|
| | Class Name | Sample Size | Class Name | Sample Size | Class Name | Sample Size |
| 1 | Water | 158 | Scrub | 761 | Alfalfa | 46 |
| 2 | Primary Floodplain | 228 | Willow swamp | 243 | Corn-notill | 100 |
| 3 | Riparian | 237 | CP hammock | 256 | Corn-mintill | 270 |
| 4 | Firescar | 178 | CP/Oak | 252 | Corn | 237 |
| 5 | Island interior | 183 | Slash pine | 161 | Grass-pasture | 59 |
| 6 | Woodlands | 199 | Oak/Broadleaf | 229 | Grass-trees | 93 |
| 7 | Savanna | 162 | Hardwood swamp | 105 | Grass-pasture-mowed | 28 |
| 8 | Short mopane | 124 | Graminoid marsh | 431 | Hay-windrowed | 478 |
| 9 | Exposed soils | 111 | Spartina marsh | 520 | Oats | 20 |
| 10 | | | Cattail marsh | 404 | Soybean-notill | 66 |
| 11 | | | Salt marsh | 419 | Soybean-mintill | 123 |
| 12 | | | Mud flats | 503 | Soybean-clean | 256 |
| 13 | | | Water | 927 | Wheat | 205 |
| 14 | | | | | Woods | 120 |
| 15 | | | | | Buildings-grass-trees-drives | 297 |
| 16 | | | | | Stone-steel-towers | 93 |

**Table 2.** Sample size (number of pixels) of each class in datasets from Group II.

| Class No. | Indiana Pine (IND PINE) | | Salinas Scene (SAL) | | Pavia University (PAV) | |
|---|---|---|---|---|---|---|
| | Class Name | Sample Size | Class Name | Sample Size | Class Name | Sample Size |
| 1 | Alfalfa | 46 | Brocoli_green_weeds_1 | 2009 | Water | 824 |
| 2 | Corn-notill | 1428 | Brocoli_green_weeds_2 | 3726 | Trees | 820 |
| 3 | Corn-mintill | 830 | Fallow | 1976 | Asphalt | 816 |
| 4 | Corn | 237 | Fallow_rough_plow | 1394 | Self-Blocking Bricks | 808 |

**Table 2.** *Cont.*

| Class No. | Indiana Pine (IND PINE) | | Salinas Scene (SAL) | | Pavia University (PAV) | |
|---|---|---|---|---|---|---|
| | Class Name | Sample Size | Class Name | Sample Size | Class Name | Sample Size |
| 5 | Grass-pasture | 483 | Fallow_smooth | 2678 | Bitumen | 808 |
| 6 | Grass-trees | 730 | Stubble | 3959 | Tiles | 1260 |
| 7 | Grass-pasture-mowed | 28 | Celery | 3579 | Shadows | 476 |
| 8 | Hay-windrowed | 478 | Grapes_untrained | 11271 | Meadows | 824 |
| 9 | Oats | 20 | Soil_vinyard_develop | 6203 | | |
| 10 | Soybean-notill | 972 | Corn_senesced_green_weeds | 3278 | | |
| 11 | Soybean-mintill | 2455 | Lettuce_romaine_4wk | 1068 | | |
| 12 | Soybean-clean | 593 | Lettuce_romaine_5wk | 1927 | | |
| 13 | Wheat | 205 | Lettuce_romaine_6wk | 916 | | |
| 14 | Woods | 1265 | | | | |
| 15 | Buildings-Grass-Trees-Drives | 386 | | | | |
| 16 | Stone-Steel-Towers | 93 | | | | |

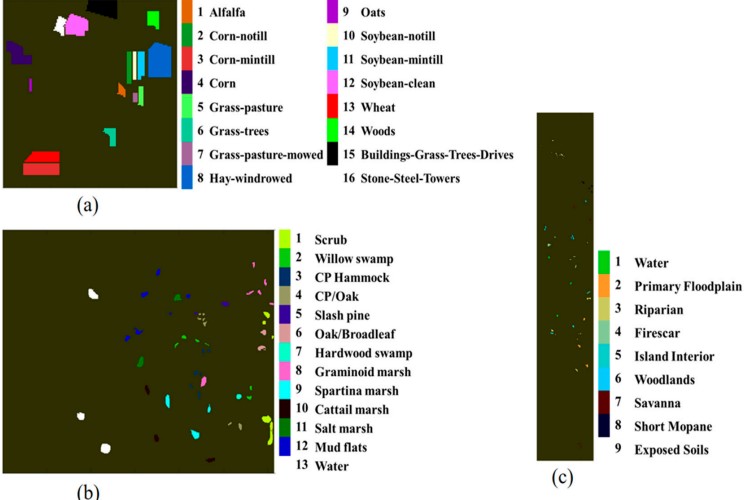

**Figure 3.** Ground truth of datasets in Group I. (**a**) Ground truth of the truncated Indian Pines (IND PINE) image. (**b**) Ground truth of the Kennedy Space Center (KSC) image. (**c**) Ground truth of the Botswana (BOT) image.

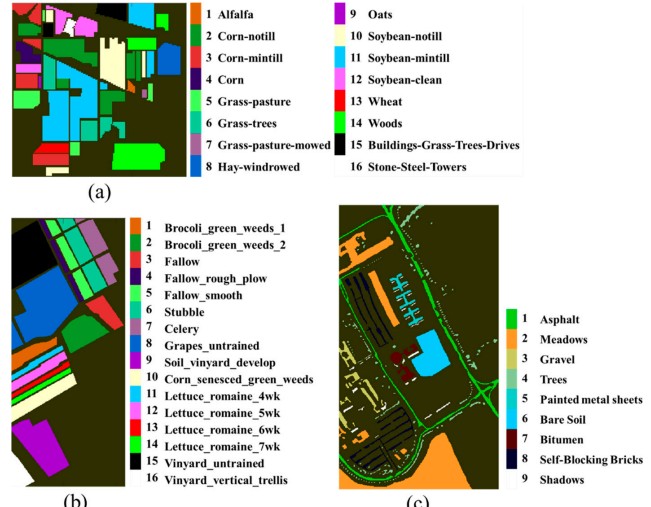

**Figure 4.** Ground truth of datasets in Group II. (**a**) Ground truth of the truncated Salinas (SAL) image. (**b**) Ground truth of the IND PINE image. (**c**) Ground truth of the Pavia University (PAV) image.

### 4.2. Experimental Setup

In this part, we evaluate the performance of our CASD assisted PCSSR algorithm on all datasets, and its performance on group I datasets will be compared to other traditional graph-based classification methods stated in [17], including the original PCSSR graph method, the Gaussian kernel (GK) graph method, the nonnegative local linear reconstruction (LLR) graph method, the local linear embedding (LLE) graph method, the nonnegative low-rank and sparse (NNLRS) graph method, and the SR graph method. Our CASD assisted PCSSR approach is implemented under the same label propagation framework as other models, and the hyperparameters from other models stay the same as [17]. The process of hyper-parameter determination during our model development will be stated in Section 4.4.

We separate every dataset into two parts, i.e., the training set and the testing set. In our case, the latter is much larger than the former. For each dataset, we randomly pick out 3/5/10/15/20 samples per class as the training set (the labeled samples), and the rest as the testing set (the unlabeled samples). An example of dividing IND PINE dataset is illustrated by Figure 5. To accord with [17], we run our algorithm 20 times for each dataset. The mean of overall accuracy (OA), average accuracy (AA), and the Kappa coefficient are utilized to evaluate the classification results.

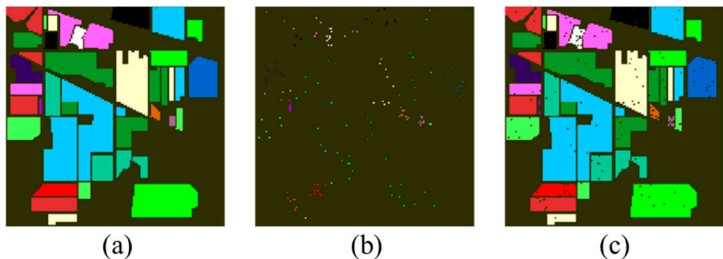

(a)                    (b)                    (c)

**Figure 5.** An example of dividing IND PINE dataset into training set and testing set. (**a**) The complete IND PINE dataset. (**b**) Training set generated by randomly picking out 10 samples per class from the complete IND PINE dataset. (**c**) Testing set.

### 4.3. Results and Discussion

Figure 6 shows how the classification overall accuracy (OA) changes with the number of labeled samples on six different datasets, and Figure 7 demonstrates the visualized classification results. For the classification result on the KSC dataset, as illustrated in Figure 6a, the CASD assisted PCSSR-graph method performs better than other methods when the number of labeled samples is more than 5 per class, finally achieving an accuracy about 97% and about 10% higher than other methods. For the result on the BOT dataset, as illustrated in Figure 6b, the CASD assisted PCSSR-graph method performs better than other methods when the number of labeled samples is more than 5 per class, finally achieving an accuracy about 99% and about 5% higher than other methods. For the result on the truncated IND PINE dataset presented in Figure 6c, the performance of our method surpasses other methods all along, and obtains an accuracy about 96% and about 16% higher than other methods when the number of labeled samples is 15 per class.



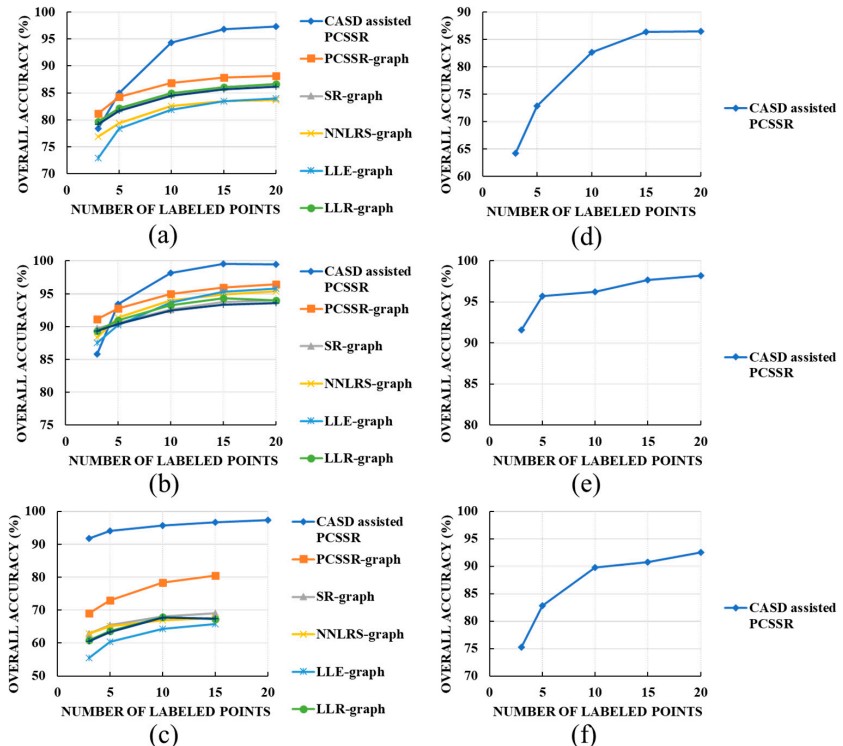

**Figure 6.** Overall accuracy with different number of labeled samples on all datasets. (**a**) KSC data with 13 classes ($\lambda_1 = 1 \times 10^{-4}, \lambda_2 = 2 \times 10^{-5}$). (**b**) BOT data with 9 classes ($\lambda_1 = 1 \times 10^{-4}, \lambda_2 = 2 \times 10^{-6}$). (**c**) Truncated IND PINE data with 16 classes ($\lambda_1 = 1 \times 10^{-4}, \lambda_2 = 2 \times 10^{-5}$). (**d**) IND PINE data with 16 classes ($\lambda_1 = 1 \times 10^{-4}, \lambda_2 = 4 \times 10^{-5}$). (**e**) SAL data with 16 classes ($\lambda_1 = 1 \times 10^{-4}, \lambda_2 = 6 \times 10^{-6}$). (**f**) Pavia University (PAV) data with 9 classes ($\lambda_1 = 1 \times 10^{-4}, \lambda_2 = 4 \times 10^{-6}$).

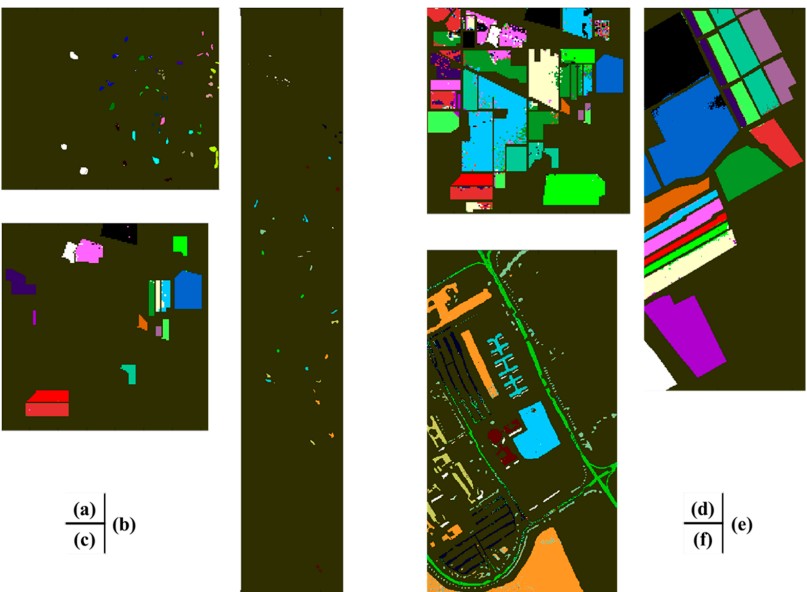

**Figure 7.** A demonstration of the typical classification results on six datasets. (**a**) KSC data with 20 labeled samples selected per class; overall accuracy (OA) = 96.16% ($\lambda_1 = 1 \times 10^{-4}, \lambda_2 = 2 \times 10^{-5}$). (**b**) BOT data with 20 labeled samples selected per class; OA = 99.93% ($\lambda_1 = 1 \times 10^{-4}, \lambda_2 = 2 \times 10^{-6}$). (**c**) Truncated IND PINE data with 15 labeled samples selected per class; OA = 98.36% ($\lambda_1 = 1 \times 10^{-4}, \lambda_2 = 7 \times 10^{-5}$). (**d**) IND PINE data with 20 labeled samples selected per class; OA = 86.68% ($\lambda_1 = 1 \times 10^{-4}, \lambda_2 = 4 \times 10^{-5}$). (**e**) SAL data with 20 labeled samples selected per class; OA = 98.72% ($\lambda_1 = 1 \times 10^{-4}, \lambda_2 = 6 \times 10^{-6}$). (**f**) PAV data with 20 labeled samples selected per class; OA = 92.68% ($\lambda_1 = 1 \times 10^{-4}, \lambda_2 = 4 \times 10^{-6}$).

Furthermore, the classification accuracy of each class, the overall accuracy (OA), the average accuracy (AA), and the Kappa coefficient for the different graph-based methods on three datasets are shown in Tables 3–5, where the highest value of each row is shown in bold. For the BOT dataset, Table 3 exhibits that our method outperforms all the other algorithms with the best class-specific accuracies on almost all indices on all classes. The only exception is that on Class Two, our method achieves an accuracy 99.93% whereas the highest accuracy is 100.00%. For the KSC dataset, Table 4 presents that our method achieves better performance than all the other algorithms on almost all indices. The only exception is that on Class 11 our method achieves an accuracy 99.64% whereas the highest accuracy is 99.70%. For the truncated IND PINE dataset, Table 5 shows that our method outperforms all the other algorithms once again with the best class-specific accuracies on almost all indices. The only exception is that on Class Eight our method achieves an accuracy 99.64% whereas the highest accuracy is 100.00%.

All the above figures and tables clarify that the classification accuracies of our model are more satisfactory than other traditional graph-based methods. Based on the above experiment results, we can come to the following conclusions:

1.  For datasets in Group I, the CASD assisted PCSSR algorithm doesn't perform so well when a small number of labeled samples are provided. However, as more labeled samples are given, our method gradually surpasses other graph-based methods, finally by more than 5% in overall accuracy. The experiment result indicates the introduction of the spatial information can effectively improve the classification accuracy of those traditional spectral-focusing algorithms when given a relatively larger training set.

2.  For datasets in Group II, our algorithm achieves super high accuracy on the SAL dataset. While for the IND PINE dataset, compared to the truncated one in Group I, the algorithm gets poorer performance on the whole IND PINE dataset than on the truncated one.

**Table 3.** Classification accuracy of each class, OA, average accuracy (AA) and Kappa coefficients for BOT data with nine classes (20 training samples for each class). The highest value of each row is shown in bold.

| Class | GK-Graph | LLR-Graph | LLE-Graph | NNLRS-Graph | SR-Graph | PCSSR-Graph | CASD Assisted PCSSR |
|---|---|---|---|---|---|---|---|
| 1 | 99.30 | 99.30 | **100.00** | 98.60 | **100.00** | **100.00** | **100.00** |
| 2 | 99.00 | 99.00 | 97.60 | **100.00** | 99.00 | 99.50 | 99.93 |
| 3 | 95.60 | 96.60 | 96.40 | 97.50 | 94.10 | 98.00 | **99.93** |
| 4 | **100.00** | 100.00 | 100.00 | 100.00 | 100.00 | 100.00 | 100.00 |
| 5 | 93.00 | 96.10 | 99.30 | 95.37 | 95.50 | 98.70 | **100.00** |
| 6 | 78.90 | 78.30 | 86.90 | 91.70 | 80.80 | 87.00 | **100.00** |
| 7 | 97.90 | 97.90 | 94.50 | 95.20 | 97.20 | 95.90 | **99.79** |
| 8 | 91.80 | 90.90 | 86.70 | 80.60 | 91.00 | 86.00 | **100.00** |
| 9 | 84.70 | 84.50 | 86.50 | 92.60 | 87.10 | 95.70 | **99.79** |
| OA | 93.36 | 93.57 | 94.64 | 95.21 | 93.86 | 95.86 | **99.71** |
| AA | 93.36 | 93.62 | 94.21 | 94.62 | 93.86 | 95.64 | **99.94** |
| Kappa | 92.47 | 92.72 | 93.93 | 94.58 | 93.04 | 95.31 | **99.68** |

**Table 4.** Classification accuracy of each class, OA, AA and Kappa coefficients for KSC data with 13 classes (20 training samples for each class). The highest value of each row is shown in bold.

| Class | GK-Graph | LLR-Graph | LLE-Graph | NNLRS-Graph | SR-Graph | PCSSR-Graph | CASD Assisted PCSSR |
|---|---|---|---|---|---|---|---|
| 1 | 87.90 | 89.60 | 96.10 | 91.10 | 91.10 | 97.20 | **99.27** |
| 2 | 88.60 | 88.60 | 90.00 | 64.20 | 87.20 | 92.20 | **100.00** |
| 3 | 75.30 | 76.80 | 64.40 | 53.40 | 76.50 | 74.00 | **99.96** |
| 4 | 51.30 | 54.50 | 42.90 | 39.50 | 53.70 | 51.10 | **100.00** |
| 5 | 56.20 | 61.40 | 40.80 | 63.50 | 54.40 | 57.10 | **100.00** |
| 6 | 35.10 | 39.20 | 41.10 | 58.00 | 47.70 | 54.70 | **99.27** |
| 7 | 55.70 | 56.10 | 63.20 | 54.80 | 58.50 | 69.40 | **100.00** |
| 8 | 84.10 | 83.60 | 81.00 | 95.40 | 82.10 | 84.10 | **99.96** |
| 9 | 89.20 | 90.40 | 91.30 | 85.60 | 91.50 | 91.50 | **100.00** |
| 10 | **100.00** | **100.00** | 99.50 | 94.30 | **100.00** | **100.00** | **100.00** |

**Table 4.** *Cont.*

| Class | GK-Graph | LLR-Graph | LLE-Graph | NNLRS-Graph | SR-Graph | PCSSR-Graph | CASD Assisted PCSSR |
|-------|----------|-----------|-----------|-------------|----------|-------------|---------------------|
| 11 | 99.10 | 99.20 | 97.00 | 75.50 | **99.70** | **99.70** | 99.64 |
| 12 | 89.70 | 91.00 | 94.80 | 89.90 | 90.00 | 90.70 | **99.64** |
| 13 | **100.00** | **100.00** | **100.00** | 99.60 | **100.00** | **100.00** | **100.00** |
| OA | 85.09 | 86.18 | 83.31 | 82.58 | 86.50 | 88.48 | **97.13** |
| AA | 77.86 | 79.26 | 77.08 | 74.22 | 79.42 | 81.67 | **99.83** |
| Kappa | 83.39 | 84.60 | 81.51 | 80.57 | 84.96 | 87.16 | **98.74** |

**Table 5.** Classification accuracy of each class, OA, AA and Kappa coefficients for truncated IND PINE data with 16 classes (15 training samples for each class). The highest value of each row is shown in bold.

| Class | GK-Graph | LLR-Graph | LLE-Graph | NNLRS-Graph | SR-Graph | PCSSR-Graph | CASD Assisted PCSSR |
|-------|----------|-----------|-----------|-------------|----------|-------------|---------------------|
| 1 | 37.40 | 33.30 | 24.30 | 41.30 | 36.10 | 73.30 | **100.00** |
| 2 | 52.10 | 51.70 | 47.50 | 36.90 | 53.50 | 79.80 | **99.69** |
| 3 | 97.20 | 97.10 | 98.40 | 83.90 | 98.10 | 99.10 | **99.11** |
| 4 | 82.90 | 84.60 | 90.60 | 98.60 | 84.20 | 95.70 | **100.00** |
| 5 | 57.90 | 59.00 | 55.00 | 96.00 | 64.30 | 60.80 | **99.87** |
| 6 | 61.70 | 59.20 | 72.40 | 51.50 | 61.80 | 86.00 | **100.00** |
| 7 | 5.40 | 6.20 | 3.20 | 40.00 | 6.40 | 10.60 | **99.87** |
| 8 | 99.60 | 99.60 | **100.00** | 98.60 | **100.00** | 99.50 | 99.64 |
| 9 | 10.20 | 9.40 | 6.80 | 10.60 | 10.00 | 11.40 | **100.00** |
| 10 | 21.60 | 24.50 | 33.30 | 39.50 | 25.00 | 37.00 | **99.51** |
| 11 | 38.70 | 41.80 | 44.70 | 36.50 | 43.50 | 64.40 | **99.47** |
| 12 | 76.90 | 79.10 | 83.20 | 88.60 | 84.10 | 94.30 | **99.47** |
| 13 | 95.70 | 95.10 | 98.90 | 90.00 | 97.80 | 98.50 | **99.11** |
| 14 | 48.70 | 46.20 | 54.00 | 39.80 | 46.50 | 51.60 | **100.00** |
| 15 | 81.70 | 83.00 | 84.20 | 95.20 | 84.70 | 90.00 | **100.00** |
| 16 | 94.10 | 95.20 | 94.00 | 39.70 | 95.20 | 93.00 | **99.47** |
| OA | 64.31 | 64.80 | 64.23 | 65.06 | 66.64 | 81.10 | **97.07** |
| AA | 60.11 | 60.31 | 61.91 | 61.67 | 61.95 | 71.56 | **99.70** |
| Kappa | 61.42 | 61.93 | 61.53 | 61.78 | 63.87 | 79.19 | **97.31** |

Conclusion 1 states that the performance of the CASD assisted PCSSR algorithm is highly related to the number of labeled samples for each class. Lack of labeled samples leads to low accuracy and the increment of labeled samples can improve the result effectively.

Since the result of the PCSSR algorithm is regularized by the probabilistic class structure which is generated by our CASD algorithm, the distances (CASDs) between samples have a great effect on the final performance of our algorithm. We can do the following operations to visualize the effect of the distances: for every unlabeled sample, find out the labeled sample with the shortest CASD to it, then mark that unlabeled sample. The classification results on BOT dataset (with three labeled samples per class) are shown in Figure 8. Please notice that "the visualization of the CASD" is an independent process, which is only for a better understanding of how well the CASD is measured. It is not an intermediate result of CASD assisted PCSSR algorithm.

It is easy to see, if the labeled samples we select from different categories are very limited, they can't be sufficiently assigned to every ground block in testing set. During the classification of such a sample block, if the samples of the same category are far away or the samples of the different classes are nearby, misclassification is likely to happen. The flaws in the probabilistic class structure generated by spatial algorithm can interfere the following sparse representation process, finally resulting in a decrease of accuracy. With the number of labeled samples increasing, the probability that a block is assigned to labeled samples will rise, the accuracy of the algorithm will be improved, and finally, the OA will be improved.

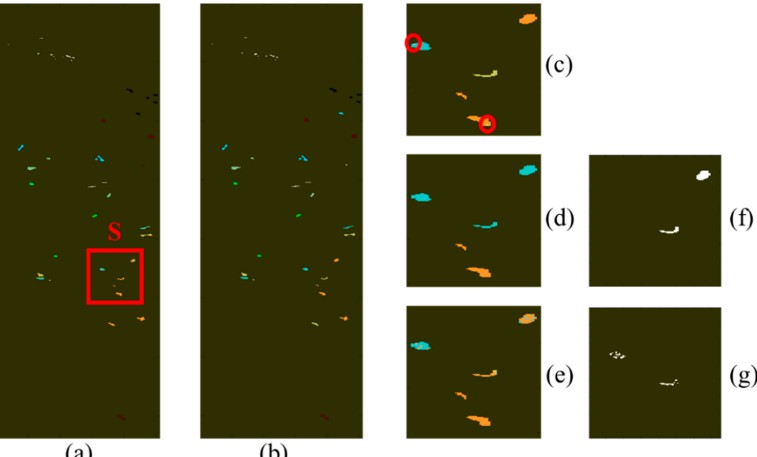

**Figure 8.** Classification result of BOT with three training samples per class (OA = 87.1%). (**a**) The classification result of BOT. (**b**) The ground truth of BOT. (**c**) The randomly selected training samples (marked with red circles) in region S. (**d**) The visualization of CASD's effect. The labels of test samples are decided by the labeled samples nearby. Several ground blocks have been misclassified. (**e**) The final classification result of the CASD assisted PCSSR algorithm. (**f**) The classification errors in (**d,g**). The classification errors in (**e**).

The spatial algorithm performs well only when the samples to predict are close to the labeled samples. As the distance increases, the reliability of prediction will drop. Besides, the classification boundary delineated by the spatial algorithm does not take into account the edge information of the hyperspectral figure. Therefore, samples in the intersecting area between classes are more affected by neighbor samples and more likely to be assigned to an incorrect category. If there are many unlabeled samples near the intersecting area, the classification result based on CASD could be unsatisfactory (Figure 9). To sum up, the classification effect will be relatively poor at the category boundary away from the training samples. Conversely, if the ground blocks to be classified in the dataset are broken and scattered, the classification boundary is more likely to fall in negligible areas (the black background area), and the classification center is more likely to fall within the ground block that needs to be classified. Therefore, with enough training samples given, the classification result on the more scattered dataset are basically better.

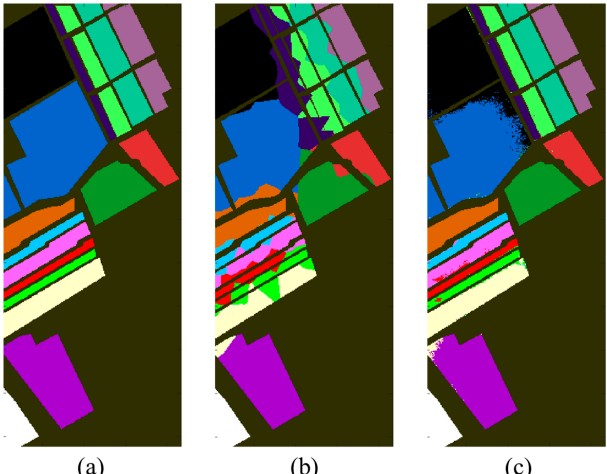

**Figure 9.** Classification result of SAL with 10 training samples per class (OA = 96.4%). (**a**) The ground truth of SAL. (**b**) The visualization of CASD's effect. The intersecting lines between classes are badly drawn. (**c**) The final classification results of CASD assisted PCSSR. Most of errors are corrected in the SR process.

### 4.4. Parameters Sensitivity Analysis

In this subsection, we will discuss the parameter sensitivity of our model using the truncated IND PINE dataset with 10 labeled samples selected from every class. There are two parameters in PCSSR algorithm, $\lambda_1$ and $\lambda_2$. $\lambda_1$ controls the sparsity of $W$ while $\lambda_2$ controls the effect of class structure regularizer. We repeat 50 runs for each fixed parameter configuration and present the average results. For example, in Figure 10a, we use a fixed $\lambda_1$ value and varies $\lambda_2$ value to observe the classification results. For each $\lambda_2$ value to be observed, we repeat 50 runs, calculate the classification accuracy during each run, and finally obtain the average accuracy. During the experiment, we first keep $\lambda_1$ equal to $1 \times 10^{-4}$ and vary the value of $\lambda_2$ from $1 \times 10^{-5}$ to $1 \times 10^{-4}$ with the step of $1 \times 10^{-5}$. As we can see from Figure 10a, the algorithm reaches the optimal performance when $\lambda_2$ equals $7 \times 10^{-5}$. Then we fix $\lambda_2$ and let $\lambda_1$ change. As illustrated in Figure 10b,c, the OA basically keeps the same when $\lambda_1$ is between $1 \times 10^{-5}$ to $1 \times 10^{-4}$, and drops when $\lambda_1$ is larger. The result shows that sparsity and probabilistic structure both matter in the classification process, though the variety of performance isn't so great when parameters change.

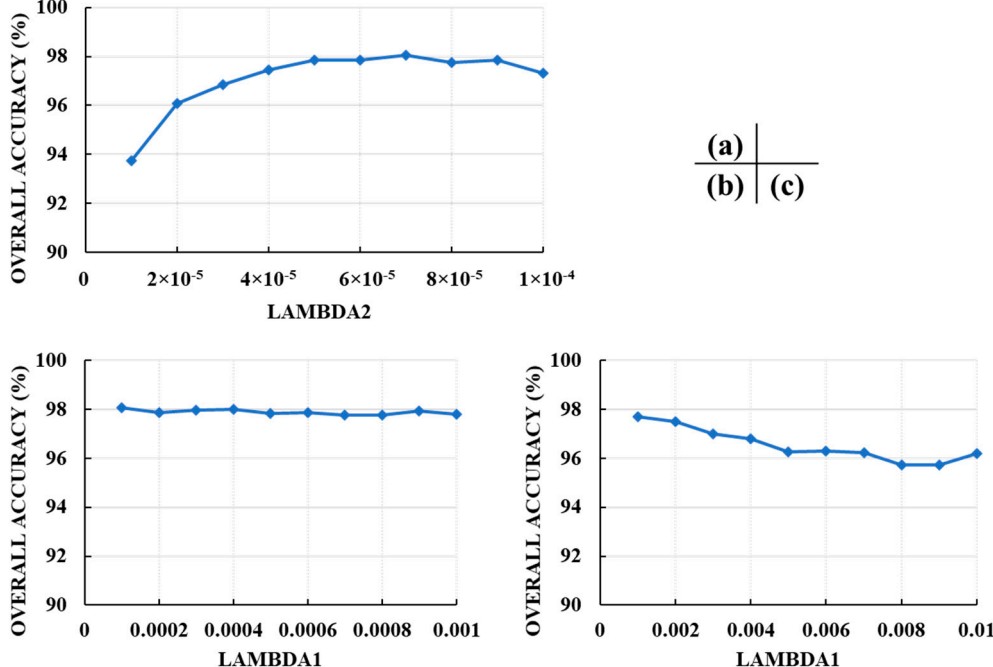

**Figure 10.** Parameter sensitivity analysis of the model. (**a**) Effect of parameter $\lambda_2$ in truncated IND PINE with 10 training samples per class ($\lambda_1 = 1 \times 10^{-4}$). (**b**,**c**) Effect of parameter $\lambda_1$ in truncated IND PINE with 10 training samples per class ($\lambda_2 = 7 \times 10^{-5}$).

### 5. Conclusions

This paper has developed a novel graph construction method called CASD assisted PCSSR algorithm. The proposed method introduces the spatial information into the classification process on the SR graph, so that the "distance" of two samples can be measures by both spatial distance and class distance. It is shown by the experimental result that CASD assisted PCSSR algorithm is an effective method for hyperspectral data classification and can achieve a relatively high performance when enough training samples are provided.

The shortage of our method also exists: Firstly, the number of training samples should be sufficient for the training process. If the training set is very limited while the ground blocks to predict are in large numbers, the final performance might be not as good. However, due to the sparse representation model used in this work, we only need a relatively small size of training set to accomplish model training. Secondly, categorizing by CASD doesn't assure a well-delineated intersection line between

classes, which means the samples close to that line might be badly classified. Nevertheless, the final output of the model could be corrected by the following sparse representation process since the CASD algorithm only provides a "suggestion" to the PCSSR algorithm. Our future work is to extract the edge information from the hyperspectral data. Applying it to the CASD algorithm may compensate for the lack of classification accuracy in the intersecting area between classes.

**Author Contributions:** Conceptualization, W.X.; methodology, W.X.; software, W.X. and S.L.; validation, W.X., S.L. and Y.W.; formal analysis, W.X.; writing—original draft preparation, W.X. and S.L.; writing—review and editing, W.X., S.L., Y.Z. and Y.W.; visualization, W.X.; supervision, Y.Z. and G.C. All authors have read and agreed to the published version of the manuscript.

**Funding:** This research was supported by the Scientific Research Training for Undergraduates of Nanjing University of Science and Technology, and partially supported by the Natural Science Foundation of Jiangsu Province under Grant BK20191284.

**Acknowledgments:** We acknowledge editors and reviewers for their valuable suggestions and corrections.

**Conflicts of Interest:** The authors declare no conflict of interest.

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
