# Peer review of "Sparse Representation Graph for Hyperspectral Image Classification Assisted by Class Adjusted Spatial Distance"

_applsci, doi:10.3390/app10217740_

Round 1
Reviewer 1 Report
Dear authors,
The article refers to known algorithm for HSI - the probabilistic class structure regularized sparse representation (PCSSR) approach and a reader would appreciate when wider context of the proposed method - CASD assisted PCSSR is explained in the introduction. Countless algorithms might be developed but at the beginning researchers must have an idea for which purpose would be a new algorithm used in comparison with existing approaches. Results of the new approach CASD assisted PCSSR were compared with results conducted with other methods: GK, LLR, LLE, NNLRS and SR but the explanation why these approaches were selected for the results comparison is missing in section “2. Related Works”. The prevailing content of the 2nd section more refers to workflow and shall be a part of section 3. Modelling and Algorithm. The authors referred to works of key authors related with proposed CASD method in the 2nd section but some definitions or explanations are not clear: for instance lines 70/79; 86-87; 89-90; 92; 99; 121-122 (see detailed notes). Further, it is not clear how the second aim was achieved: “the performance of the PCSSR algorithm by 5% to more than 10% when enough training samples are provided“.
Data from the presented localities were used also in other works referring to HSI classification performance, I think some reference to this work/s has to be mentioned in the article. The method that the authors developed is only one of several and each method has its limits of the application. This aspect is not introduced and neither discussed in the article. Therefore, the interpretation of results has to be given to a wider context of methods which were used for data comparison (GK, LLR, LLE, NNLRS and SR). Otherwise at this stage it is difficult to say if the results are sufficient to classify all categories of land cover in various spatial relations and also to prove that CASD exactly improved the performance of PCSSR algorithm.
Detailed comments (lines) are bellow and highlighted text in the attached PDF: applsci-948804-peer-review-v1
- 25: add two other indicative key words because three do not sufficiently characterize the article: for instance “regularizer” “spatial distance information” ... or whatever characterizing the specificity of the proposed method.
- 35-36: insert references after each method
- 54-55: Explain the usage of HSI classifications, there exist a numerous implementations, e.g.: A review on graph-based semi-supervised learning methods for hyperspectral image classification / Shrutika S.Sawant, Manoharan Prabukumar but at the beginning the explanation why a new approach was developed would be expected. The basic question has to be answered – did you find out any difficulties with a particular HSI applications (depending on land cover specifics ...). Many different land cover categories were tested in this article and at the end the authors noted: line 300-304 that some categories are more difficult to recognize that other ones using CASD assisted PCSSR. It is normal that each method would fit better/worse to a certain land cover categories; maybe some categories in a certain spatial relations are indifferent to classification whichever method is used. At the beginning an assumption has to be set up – for which purpose did you plan to use a new CASD algorithm?
- Aims: Did you compared you approach with other works? For instance, Wu et al., 2020 /Semi-Supervised Hyperspectral Image Classification via Spatial-Regulated Self-Training: the authors defined one of their aim „Adjacent pixels in a hyperspectral image may belong to the same class. We introduce a spatial constraint in the above algorithm to give a smoothness hypothesis to improve HSI accuracy“
- 61: Explain why did you chose performance values/thresholds to 5% and 10% and where a reader can be find the verification of this assumption in results?
- 75: “V” are nodes and “E” are .... edges?
- 70/79: Explain using more words or sentences the context of the defined aims and “a problem the minimizing harmonizing energy” is not clear. Surely, the authors Zhu, X.; Ghahramani, Z.; Lafferty, J.D. Semi-supervised learning using gaussian fields and harmonic functions published the article that is cited in line 70 but relationship between minimizing of harmonic energy is SSL and the proposed concept of the class adjusted spatial distance between pairs of samples has remained unclear.
- 79: is a formula adopted from Zhu eta l. 2003?
- 86-87: I do not understand the sentence: “Since we use the LP algorithm to propagate the ...“
- 89-90: The development of methods for the creation of and adjacency matrix must be defined in the aim following line 51: “We propose the concept of the class adjusted spatial distance …”
“R” is regularizer and in line 115 R (W) it is defined as a graph edge matrix - OK. But what function has “R” in formula – line 90 where the adjacency matrix of the graph is defined?
R (W) = write exactly is it „the adjacency matrix of the graph” or “a graph edge matrix”
- 92: “introduce a probabilistic class structure regularizer into the SR based model to increase its discriminability” : it must be commented in the proposed objectives (lines 55-62) including reference no. 12.
- 99: This sentence makes no sense.
- 121-122: How can ADMAP help to solve PCSSR model? It is stated at the end of this section but the answer is expected if the authors considered to refer to ADMAP solution. Otherwise, remove it.
- Abstract refers (line 16) to approach “the probabilistic class structure regularized sparse representation (PCSSR) approach“ :, in lines 121-122 PCSSR is called model– but model is exactly related to a certain case, situation, object ... it shall be only SR model developed by PCSSR approach or not? Explain.
- 129-130: The proposed PCSSR approach is valid only for the calculation of short Euclidian distances on a geoid. The authors mentioned distances amongst land cover categories and also a case when the algorithm fails (line 135-136). Maybe it would be useful to explain that the calculation of distances on the geoid works differently than on the Cartesian plane and consequently to continue to 3.2 explaining how CASD copes with this problem.
- 159, Figure 1: the shortest path “the shortest path between A and C is marked in green but green is C-F.
- 204-206: „the gaussian kernel (GK) graph method, the nonnegative local linear reconstruction (LLR) graph method, the local linear embedding (LLE) graph method, the nonnegative low-rank and sparse (NNLRS) graph method, and the SR graph method“ : all of these methods had to be introduced in section 2. Related work and a purpose why the selected algorithms: GK, LLR, LLE, NNLRS and SR were used for the data comparison must be explained in this section.
- Discussion: and what about other approaches which were performed at the same localities: http://www.ehu.eus/ccwintco/index.php/Hyperspectral_Remote_Sensing_Scenes (for instance: Representative Band Selection for Hyperspectral Image Classification August 2018 International Journal of Geo-Information 7(9): 338 DOI: 10.3390/ijgi7090338)
- 240-245: I do not understand the content of these sentences, explain in detail why only accuracies values “8 of 9 ”; “12 of 13”; “15 of 16” were acceptable but 99,64 not.
- Line 284: “c) The final classification results. Most of errors are corrected in the SR process.“ Did you mean a SR process to compute similarity information of which SR process? Because in previous image is visualized CASD’s effect. Which SR process was used in c)? and consequently it shall be explained in context to the sentence in line 335 „....,which means the samples close to that line might be badly classified.“ But if a SR process was used to decrease misclassification it shall be mentioned also in conclusions. Some very short conclusions on SR process are in line 338-340 but it requires deeper explanation.
regards, Reviewer

Reviewer 2 Report
Dear Authors,
This is an interesting study of a new graph construction method that adds spatial information into the classification process on the SR graph. Authors use five different hyperspectral datasets divided in two groups to evaluate the proposed algorithm. Overall, this manuscript is clearly written and informative. The paper is thus worth to be published after the minor suggestions listed below.
- The abstract needs rewriting: please add the specific results to the abstract
- The introduction needs rewriting. Please enrich scientific literature using recent publications.
- Missing clear statement what is the aim of this study and objectives or research questions.
Reviewer 3 Report
The paper provides sufficient background and includes all relevant references. The research is designed appropriate, and the methods adequately are described. Anyway the use of a flowchart to explain and summarize all the research should improve the quality of the paper.
The results are clearly presented, but authors could add the image to be analyzed. The conclusions are supported by the results
Round 2
Reviewer 1 Report
Dear authors,
I appreciate your effort; the article was markedly improved - mainly the importance of a new approach using the distance information in PCSSR algorithm. The main problem with the first review was misunderstanding at my side, but I told to myself that it is better to ask although maybe irrelevant questions like to leave it as it is with a possible mistake.
I am fully satisfied with exhaustive explanation and detailed clarification which are well written. I can say that you spent a lot of time revising the manuscript and its quality and interest to readers is at high level now.
I suggest to accept the article in present form. I hope that similar opinion/s will have other reviewer/s,
regards Reviewer
This manuscript is a resubmission of an earlier submission. The following is a list of the peer review reports and author responses from that submission.
Round 1
Reviewer 1 Report
This manuscript reports hyperspectral image classification by class adjusted spatial distance using sparse representation graph. The aim of this paper is clear and contribution of research benefits to readers who conduct hyperspectral data analysis. Although the paragraphs are concise, missing information makes the manuscript difficult to understand so this paper needs major revision to improve the quality for better understanding. More specific questions are as follows:
Line 80: how did you develop well-construct graph?
Line 81: how did you find good and proper method? with what data? how do we know an optimum?
Line 86: do you have specific types of corrupted and noise data?
Line 95-96: what is the advantage of your algorithm compare to a Mahalanobis distance method?
Line 97: how did you determine this number in real samples?
Line 100: how far between two samples?
Line 118-119: any data preprocessing could help recover this issue?
Line 125: need a space after “V”
Line 188: how did you determine hyperparameters for model development?
Line 189: it is not clear how to split datasets as training and testing.
Line 190: what is pixel size of each sample?
Line 192: did you run 20 times for all samples regardless of the sample size?
Line 193: what's the difference between OA and AA in this study?
Line 209: show examples of other graph-based methods.
Line 211: show example how effectively improve classification accuracy.
Line 235: mark misclassification areas for better understanding in Figure 5.
Line 265-266: explain this more detail. Do you mean 50 runs with different parameter values?
Line 282: Does this mean large size of ground truth data is required for training? If not, is this algorithm not working well? Clarify this claim.
Reference #6, #13: missing year
Reviewer 2 Report
In the introduction, authors state that "The main contributions of this paper are two folds: [..] a method to estimate the similarity information needed in the PCSSR algorithm " and the "concept of the CASD". It is not clear where the first contribution is described in the paper, is it the "Label Propagation Algorithm"? If so, it should not be in the "related work" section; if not, authors should better highlight this contribution in the paper (maybe dedicating a specific section to it).
Editorial comments:
- At line 125, correct "Vrepresents" with "V represents".
